# Indole-3-Butyric Acid Enhances Root Formation and Alleviates Low-Temperature Stress in Sugarcane: Molecular Insights and Identification of Candidate Genes

**DOI:** 10.3390/plants14101502

**Published:** 2025-05-16

**Authors:** Xiao-Qiu Zhang, Yong-Jian Liang, Xiu-Peng Song, Mei-Xin Yan, Li-Qiu Tang, Zhen-Qiang Qin, Yu-Xin Huang, De-Wei Li, Dong-Mei Huang, Ze-Sheng Shi, Bao-Qing Zhang, Dong-Liang Huang

**Affiliations:** 1Guangxi Key Laboratory of Sugarcane Genetic Improvement, Key Laboratory of Sugarcane Biotechnology and Genetic Improvement (Guangxi), Sugarcane Research Institute, Guangxi Academy of Agricultural Sciences, Nanning 530007, China; zhangxiaoqiuxhd@163.com (X.-Q.Z.); yongjianliang_605@163.com (Y.-J.L.); xiupengsong@163.com (X.-P.S.); yanmeixin@gxaas.net (M.-X.Y.); qinzqcn@163.com (Z.-Q.Q.); huangyuxin13@163.com (Y.-X.H.); ldw11023@163.com (D.-W.L.); huang18260951004@163.com (D.-M.H.); 2Guangxi South Subtropical Agricultural Science Research Institute, Chongzuo 532415, China; tangliqiu_726@163.com (L.-Q.T.); shizesh@126.com (Z.-S.S.)

**Keywords:** abiotic stress, root system, phytohormone, transcriptome, seedcane

## Abstract

Sugarcane (*Saccharum officinarum* L.) faces significant challenges in China, including labor-intensive cultivation, low yields, and environmental stresses. Enhancing root development and stress tolerance through phytohormones and molecular breeding is a promising approach to boosting productivity. Indole-3-butyric acid is a phytohormone known for promoting root development and stress resistance. However, its effects on sugarcane root development under low temperature remain poorly understood. This study demonstrated that IBA markedly promoted root initiation, elongation, and biomass under low temperature, and significantly increased the levels of phytohormones, including GA_3_, ABA, JA, IAA, and ZT, suggesting the activation of multiple signaling pathways. Transcriptome analysis revealed numerous differentially expressed genes related to metabolic pathways such as glycolysis, the tricarboxylic acid cycle, and glutathione metabolism. Weighted gene co-expression network analysis identified core gene modules correlated with phytohormone activities, highlighting their role in the IBA-mediated stress response. Eleven core genes, including *GSTU6*, *FAR1*, and *BCAT3*, and nine hub genes, such as *Ub-CEP52-1* and *ACS1*, were identified as critical components for IBA-induced root development and stress mitigation. These findings provide insights into the molecular mechanisms underlying IBA-induced root development and stress tolerance in sugarcane, offering candidate genes for breeding high-yield, stress-tolerant varieties and demonstrating IBA’s potential as a strategy to enhance productivity under challenging conditions.

## 1. Introduction

Sugarcane (*Saccharum officinarum* L.) is the leading global crop for sugar production and the second-largest source of bioenergy. In China, it is the primary sugar crop, primarily cultivated in southern and southwestern regions. However, sugarcane production in China faces significant challenges, including labor-intensive cultivation practices, low yields, and susceptibility to environmental stresses, such as low temperature [1]. Addressing these issues requires integrated strategies such as improved cultivation methods and the development of stress-tolerant varieties.

The root system is essential for water and nutrient uptake, transport, and storage, playing a pivotal role in supporting sugarcane growth [2]. An extensive root system enhances the plant’s capacity to access soil resources, thereby improving productivity [3]. Indole-3-butyric acid (IBA), a precursor of the auxin indole-3-acetic acid (IAA), is converted through peroxisomal β-oxidation. As a key phytohormone, IBA is widely used to promote root formation in numerous crops [4,5]. In sugarcane, IBA application induces adventitious root development by increasing root number and length during micropropagation [6] and enhances root establishment following transplantation [7]. Seedcane soaking in IBA solutions further stimulates root growth, increasing root length and biomass, which contributes to greater plant height and stem diameter [8].

Beyond root development, IBA also contributes to abiotic stress mitigation. It enhances antioxidative defense mechanisms, nitric oxide production, glutathione peroxidase activity, polysaccharide accumulation, and cadmium-binding capacity, thereby alleviating cadmium toxicity and drought stress [9,10,11]. Under phosphate deprivation, IBA homeostasis and conversion to IAA are modulated by glutathione [12]. IBA also reduces the adverse effects of salt stress, particularly at lower salinity levels [13], and its combined application with silicon nanoparticles has been shown to mitigate oxidative stress caused by chromium toxicity [14]. These findings underscore IBA’s potential as a critical agronomic input to enhance root formation and stress resilience in sugarcane.

In parallel, the development of high-yielding and stress-tolerant sugarcane varieties is essential for overcoming production constraints in China. Identifying genes associated with root development and stress responses is a key step toward the molecular breeding of elite cultivars. In this study, we investigated the effect of IBA on sugarcane root formation under low-temperature conditions. Furthermore, we identified candidate genes involved in root development and low-temperature stress tolerance through transcriptomic and weighted gene co-expression network analysis (WGCNA). This work not only advances our understanding of the use of IBA to enhance sugarcane growth, but also provides molecular targets for the development of high-yielding, cold-tolerant varieties through biotechnological approaches.

## 2. Results

### 2.1. IBA Promoted the Development of the Root System in Sugarcane Under Low Temperature

Compared to the control, root primordia in the IBA-treated group exhibited a larger size after 15 days of incubation (Figure 1A), and root elongation was markedly enhanced after 30 days (Figure 1B). Additionally, the root fresh weight in the IBA-treated group was significantly increased (Figure 1C). These results demonstrate that IBA promotes the initiation of root primordia and facilitates root development in sugarcane under low-temperature conditions.

### 2.2. IBA Enhanced the Activities of Phytohormones in Sugarcane Roots Under Low Temperature

The activities of GA_3_, ABA, JA, IAA, and ZT in the IBA-treated group remained consistently higher than those observed in the control group across all time points, exhibiting a consistent temporal pattern: levels gradually increased from day 1 to day 7, peaked at day 7, and subsequently declined by day 9 (Figure 2). The increased levels of these five phytohormones suggest that IBA triggers the activation of multiple phytohormone signaling pathways in sugarcane roots under low-temperature conditions.

### 2.3. IBA Induced a Significant Number of Responsive Genes in Sugarcane Roots Under Low Temperature

Differentially expressed genes (DEGs) were identified based on the criteria of |fold change| > 2 and a false discovery rate (FDR) < 0.05. After 1 day of IBA treatment, 7964 DEGs were identified, including 3629 upregulated and 4335 downregulated genes. After 3 days of treatment, 6983 DEGs were identified, comprising 3061 upregulated and 3922 downregulated genes. After 5 days of treatment, 1892 DEGs were identified, including 721 upregulated and 1171 downregulated genes. After 7 days of treatment, 4443 DEGs were identified, with 2165 upregulated and 2278 downregulated genes. After 9 days of treatment, 4556 DEGs were identified, comprising 2230 upregulated and 2326 downregulated genes. Overall, the number of DEGs followed a dynamic pattern, with an initial peak at 1 day, a decline at 3 days, a minimum at 5 days, and a subsequent increase at 7 and 9 days (Figure 3A). These results demonstrate that IBA elicits widespread transcriptional responses in sugarcane under low-temperature stress.

In total, 16,754 non-redundant DEGs were identified across all comparisons. Among these, 593 DEGs were commonly expressed across all time points (Appendix A). Additionally, 3536, 1800, 164, 913, and 858 DEGs were specifically expressed at 1, 3, 5, 7, and 9 days of treatment, respectively (Figure 3B). KEGG analysis of these 16,754 DEGs revealed their involvement in pathways including glycolysis/gluconeogenesis (ko00010), the citrate cycle (TCA cycle) (ko00020), pentose phosphate metabolism (ko00030), pentose and glucuronate interconversions (ko00040), and fructose and mannose metabolism (ko00051) (Figure 3C). This suggests that IBA treatment modulates gene expression related to these biological processes.

### 2.4. Phytohormone Activities Exhibited a Strong Correlation with Gene Expression Levels in Sugarcane Roots

To identify genes associated with IBA stimulation, WGCNA was performed on the expression profiles of DEGs at different time points. These genes were grouped into 20 co-expressed modules, and their pairwise correlations were evaluated. Genes within each module showed high correlation (Figure 4A). The 20 modules were clustered into two groups with a high degree of interaction connectivity (Figure 4B). Different modules contained varying numbers of genes, with the cyan (7208 genes), turquoise (5251), brown4 (2977), floralwhite (2635), and green (2597) modules containing the most genes. The five modules with the lowest numbers of genes were grey (13 genes), darkseagreen4 (52 genes), honeydew1 (58 genes), lavenderblush3 (65 genes), and lightpink4 (67 genes) (Figure 4C). The cyan, brown4, and yellowgreen modules were significantly correlated with ABA, JA, IAA, ZT, and GA_3_, while the darkorange2 module also showed strong correlations with ABA, JA, ZT, and GA3 (Figure 4D). This suggests that the brown4, cyan, yellowgreen, and darkorange2 modules are likely key participants in IBA stimulation under low-temperature conditions and were thus identified as core modules involved in IBA-mediated responses.

Further analysis revealed that the core gene modules related to phytohormone activity displayed diverse temporal expression profiles in response to IBA treatment under low-temperature conditions (Figure 4E). Gene expression in the yellowgreen module gradually declined from day 0 to 3, peaked at day 5, and then decreased again from day 7 to 9. Expression in the brown4 module progressively increased from day 0 to 3, slightly declined at day 5, and rose again from day 7, peaking at day 9. Cyan module expression decreased at days 1 and 3, increased at day 5, and declined again at days 7 and 9. In the darkorange2 module, expression declined at day 1, rose at day 3, and then decreased again at days 5 and 9.

### 2.5. The Core Genes Associated with IBA Stimulation in Sugarcane Roots Under Low-Temperature Conditions Were Identified

Venn diagram analysis was conducted using the 593 overlapping DEGs (Figure 3B) and the brown4, cyan, yellowgreen, and darkorange2 modules. A total of 4 genes from the brown4 module and 42 genes from the cyan module were identified as common genes, representing core genes responsive to IBA stimulation in sugarcane roots under low-temperature conditions (Figure 5A; Appendix A). KEGG pathway analysis of these 46 common genes revealed enrichment in 40 metabolic pathways, of which 6 were significantly enriched. These included glutathione metabolism; cutin, suberine, and wax biosynthesis; valine, leucine, and isoleucine degradation; peroxisome; glycosphingolipid biosynthesis (globo and isoglobo series); and glucosinolate biosynthesis (Figure 5B). Based on these pathways, 11 candidate genes—*CICDH*, *GSTF11*, *GSTU6*, *GST1*, *GST4*, *FAR1*, *FAR2*, *FAR5*, *BCAT3*, *HIBCH*, and *AGAL*—were identified as putatively involved in IBA-induced responses under low-temperature conditions (Figure 5C; Appendix A).

To investigate gene-to-gene interactions among the 46 core genes, a protein–protein interaction (PPI) network was constructed using the STRING database (http://string-db.org (accessed on 14 August 2024)). The resulting network included 71 gene pairs, with 30 pairs exhibiting a combined interaction score greater than 400 (Figure 5D; Appendix A). In the network, nodes represent genes, and edges represent predicted functional associations. Based on network topology, the top nine hub genes—*Ub-CEP52-1*, *PRL19A*, *RPS28*, *RPS25B*, *ACS1*, *RPP3A*, *RPS3C*, *ACC*, and *GSTU6*—were identified as key candidate genes potentially associated with the IBA response under low-temperature conditions (Figure 5D).

### 2.6. qRT-PCR Validation Confirmed the Expression Patterns of Core Genes in Response to IBA Stimulation Under Low Temperature

To validate the accuracy of RNA-seq-derived gene expression data, nine hub genes with high connectivity in the co-expression network were selected for quantitative real-time PCR (qRT-PCR) analysis. The qRT-PCR results showed expression patterns consistent with those obtained from RNA-seq, thereby confirming the reliability of the transcriptomic data (Figure 6).

## 3. Discussion

The development of the root system is tightly regulated by auxin signaling. Indole-3-butyric acid (IBA), a widely used auxin analog and effective rooting agent, has been shown to significantly promote root system development and enhance tolerance to abiotic stresses, often exhibiting greater rooting efficacy than indole-3-acetic acid (IAA) [4,10]. In the present study, sugarcane plants treated with IBA exhibited earlier initiation of root primordia and significantly higher root fresh weight compared to the control group (Figure 1), confirming the positive effect of IBA on root growth under low-temperature conditions. Previous studies have demonstrated that the accumulation of phytohormones such as abscisic acid (ABA), gibberellic acid (GA_3_), jasmonic acid (JA), IAA, and zeatin (ZT) contributes to improved cold tolerance and enhanced plant growth [15,16,17,18]. In this study, IBA treatment led to increased levels of GA_3_, ABA, JA, IAA, and ZT, which was associated with enhanced root development under low-temperature stress. These five phytohormones were therefore selected for weighted gene co-expression network analysis (WGCNA), which revealed strong correlations between hormone levels and differentially expressed gene (DEG) expression patterns.

IBA has been reported to regulate a range of physiological processes, including lignin biosynthesis, peroxidase activity, nitric oxide (NO) production, glutathione peroxidase activity, polysaccharide content, and cadmium-binding capacity, thereby enhancing plant resilience to abiotic stress [9,10,11,19]. In the present study, the responsive genes were found to be associated with these key stress response pathways, including lignin metabolism, NO signaling, peroxidase function, lipopolysaccharide modification, glutathione peroxidase activity, and cadmium transport and tolerance. Furthermore, WGCNA revealed six core metabolic pathways responsive to IBA treatment under low-temperature conditions, highlighting the multifaceted role of IBA in regulating stress-responsive gene networks in sugarcane.

Under abiotic stress, IBA regulates plant root system development through a glutathione-dependent mechanism. Glutathione, a crucial thiol redox regulator, participates in various cellular processes by conjugating with other molecules via the glutathione S-transferase (GST) enzyme family or acting as an electron donor in antioxidant systems, thereby facilitating detoxification, sulfur homeostasis, ROS homeostasis, and redox signaling [20]. Previous studies have shown that IBA stimulation elevates glutathione peroxidase activity and the expression of glutathione metabolism-related genes [10,21]. Glutathione deficiency can disrupt root development [12]. Notably, IBA levels were found to be positively correlated with the activities of glutathione reductase (GR), glutathione S-transferase (GST), and glutathione peroxidase (GPX) under low-temperature stress [22]. In this study, genes involved in glutathione metabolism, such as *CICDH*, *GSTF11*, *GSTU6*, *GST1*, and *GST4*, were significantly downregulated in sugarcane, which contrasts with findings in barley and *Camellia sinensis* [10,21]. This suggests that the regulatory mechanism of IBA in glutathione-dependent pathways in sugarcane may be distinct from other plant species.

Cutin, suberin, and wax are key constituents of plant surface lipids that contribute significantly to the formation of epidermal barriers, playing critical roles in limiting water loss and protecting against environmental stresses. The biosynthesis of these components is closely associated with fatty alcohols, which are believed to directly mediate plant responses to abiotic stress [23]. Fatty alcohol synthesis is catalyzed by fatty acyl-CoA reductases (FARs), which are typically involved in the production of cuticular waxes in epidermal cells. FAR genes are known to respond positively to abiotic stress: their overexpression has been shown to reduce reactive oxygen species (ROS) accumulation, increase primary alcohol content, decrease cuticle permeability, and enhance stress tolerance [24]. Moreover, FARs are specifically expressed during root development and are induced by cold stress and abscisic acid [24,25,26]. In this study, *FAR1*, *FAR2*, and *FAR5* were found to be downregulated under IBA treatment in sugarcane roots, suggesting a distinct regulatory mechanism of FARs in IBA-mediated stress alleviation in sugarcane.

The branched-chain amino acid aminotransferase (BCAT) gene plays a pivotal role in branched-chain amino acid (BCAA) metabolism, plant stress responses, and glucosinolate (GSL) biosynthesis [27,28,29]. For instance, *TaBCAT1* in wheat modulates susceptibility to rust disease, while BCAT1 in rice and tomato enhances drought tolerance and leucine synthesis under stress conditions [30,31,32,33]. In this study, BCAT3 expression was significantly upregulated in sugarcane roots under low-temperature conditions following IBA treatment, suggesting its involvement in enhancing cold tolerance. Furthermore, BCAT3 is known to participate in the initial steps of methionine-derived aliphatic GSL biosynthesis, which plays a role in regulating auxin homeostasis and bolstering plant defenses against pathogens and herbivores [28,29,34,35]. The observed upregulation of BCAT3 under combined IBA and cold stress likely facilitates GSL biosynthesis, promoting indole-3-acetic acid (IAA) production and thereby enhancing root development and resistance to biotic stress [28,36].

HIBCH (3-hydroxyisobutyryl-CoA hydrolase) is a key enzyme in the L-valine catabolic pathway, functioning to prevent the accumulation of cytotoxic intermediates and playing vital roles in symbiotic nitrogen fixation and cold stress signaling [37,38,39]. HIBCH expression is upregulated in coconut under cold stress, contributing to enhanced cold tolerance [40]. However, its function in sugarcane has been rarely explored. In this study, HIBCH expression in sugarcane roots was significantly induced by IBA treatment under low-temperature conditions, suggesting that HIBCH may contribute to improved stress tolerance in sugarcane by modulating valine metabolism under abiotic stress conditions.

The conversion of IBA to IAA, along with IBA metabolism, predominantly takes place in peroxisomes. Isocitrate dehydrogenase (ICDH) is recognized as a cold-adapted enzyme [41], with NADP^+^-dependent ICDH isoforms localized in the cytosol, chloroplasts, mitochondria, and peroxisomes. Among these, the cytosolic isoform (CICDH) is closely linked to glutamate biosynthesis, and its enzymatic activity is typically enhanced under low-temperature stress [42]. CICDH deficiency has been associated with increased global trimethylation of histone H3 lysine 4 (H3K4me3), intensifying mutation defects in the H3K4me3 demethylase gene JMJ14 [43]. In this study, ICDH expression in sugarcane roots was found to be downregulated under low-temperature conditions, potentially as a result of IBA induction. Furthermore, CICDH was enriched across several metabolic pathways—including glutathione metabolism, peroxisome function, 2-oxocarboxylic acid metabolism, and amino acid biosynthesis—highlighting its critical role in mediating sugarcane root responses to IBA under cold stress.

Alpha-galactosidases (α-GALs) are glycoside hydrolases that catalyze the removal of terminal α-D-galactosyl residues from raffinose family oligosaccharides (RFOs). In plants, α-GALs play key roles in the mobilization of storage polysaccharides, particularly during seed germination and tuber sprouting [44]. Additionally, α-GALs are actively involved in phloem unloading and assimilate partitioning in sink tissues of plants that translocate raffinose. α-GALs degrade RFOs by hydrolyzing terminal galactose residues, aiding in the remobilization of stored carbohydrates [45]. These enzymes are broadly expressed across various plant tissues, with pronounced activity in the root epidermis, endodermis, and the Casparian strip [44]. In the present study, the α-GAL gene (AGAL1) was significantly upregulated under IBA treatment, indicating that IBA may enhance α-GAL activity. This upregulation likely accelerates phloem and cell wall development, thereby promoting root system growth.

The construction of a protein–protein interaction (PPI) network using the STRING database (http://string-db.org) provided crucial insights into gene interaction dynamics. This high-confidence interaction network underscored the importance of connectivity among genes in regulating sugarcane responses to IBA under low-temperature conditions. Notably, nine hub genes were identified as central nodes within the network: *Ub-CEP52-1* (ubiquitin-60S ribosomal protein), *PRL19A* (60S ribosomal protein L19-like protein), *RPS28* (40S ribosomal protein S28), *RPS25B* (40S ribosomal protein S25-2), ACS1 (ACC synthase 1), *RPP3A* (60S acidic ribosomal protein P3), *RPS3C* (40S ribosomal protein S3-1), *ACC1* (acetyl-CoA carboxylase 1), and *GSTU6* (glutathione S-transferase U6). These hub genes are implicated in essential biological processes such as stress signaling, protein biosynthesis, and metabolic regulation—key components of plant adaptive responses to abiotic stress [46,47,48,49,50]. The identification of these candidate genes offers a valuable foundation for further functional characterization aimed at deciphering their precise roles in low-temperature stress tolerance. Future studies should prioritize experimental validation of these interactions and investigate their regulation under diverse environmental scenarios, with the goal of enhancing stress resilience in crop species.

## 4. Materials and Methods

### 4.1. Plant Materials and Treatments

Sugarcane variety GT44 is a hybrid cultivar widely cultivated in major sugarcane-growing regions in China. Single-bud seedcanes, approximately 5 cm in length, were immersed in an IBA solution (60 mg/L) and then incubated at a constant temperature of 15 °C under a 16 h light/8 h dark photoperiod with a light intensity of 2000 Lux. The seedcanes were maintained in a fixed location throughout the experiment. The treated seedcanes were planted in pots (7 cm × 8 cm, diameter × height) after immersion, with one seedcane planted per nursery pot. There were 24 pots in total for each treatment. The IBA suspension was poured into the nursery pots to infiltrate the substrate prior to incubation. The nursery pots were covered with plastic film to maintain substrate moisture. After incubating for 15 days, three root points from each treatment were randomly excised and prepared as freehand sections using a microblade for optical microscopy observation. Subsequently, the IBA solution was reapplied to infiltrate the substrate 30 days after the first application. Water treatment was used as the control. Roots were collected at 0, 1, 3, 5, 7, and 9 days after the second application. During sampling, the setts were carefully removed from the substrate, and three roots were randomly collected from each sett at each time point. The setts were then replanted in the substrate and watered. The collected roots were mixed, cleaned, weighed, and immediately frozen in liquid nitrogen, and then stored at −80 °C for phytohormone activity assays and RNA-Seq analysis.

### 4.2. Phytohormone Activity Analysis

The phytohormones were extracted according to the method described by Li et al. [51] with slight modifications. A 0.5 g portion of root tissue was ground in an ice-chilled mortar and pestle with 2 mL of 80% (*v*/*v*) methanol containing 1 mM butylated hydroxytoluene as an antioxidant. The homogenate was incubated at 4 °C for 4 h and then centrifuged at 4000 rpm for 15 min at 4 °C. The resulting supernatant was collected and passed through a Sep-Pak C18 cartridge (Waters, Milford, MA, USA) for purification. The eluate was subsequently dried using a freeze dryer (Labconco, England) and reconstituted in 1 mL of phosphate-buffered saline (PBS) containing 0.1% (*v*/*v*) Tween 20 and 0.1% (*w*/*v*) gelatin (pH 7.5). Phytohormone contents—including gibberellin (GA_3_), abscisic acid (ABA), jasmonic acid (JA), indole-3-acetic acid (IAA), and zeatin (ZT)—were quantified using Enzyme-Linked Immunosorbent Assay (ELISA) kits (Michy Biology, Suzhou, China) according to the manufacturer’s protocols. All assays were conducted with six biological replicates.

### 4.3. RNA Extraction, Library Construction, and Sequencing

This procedure was performed following the protocol provided by Gene Denovo Biotechnology Co. (Guangzhou, China). Total RNA was extracted using the Trizol reagent kit (Invitrogen, Carlsbad, CA, USA) following the manufacturer’s protocol. RNA quality was assessed on an Agilent 2100 Bioanalyzer (Agilent Technologies, Palo Alto, CA, USA) and verified using RNase-free agarose gel electrophoresis. After total RNA extraction, mRNA was enriched using Oligo(dT) beads. The enriched mRNA was fragmented into short fragments using fragmentation buffer and reverse-transcribed into cDNA using the NEBNext Ultra RNA Library Prep Kit for Illumina (NEB #7530, New England Biolabs, Ipswich, MA, USA). The purified double-stranded cDNA fragments were end-repaired, and an A base was added and ligated to Illumina sequencing adapters. The ligation reaction was purified using AMPure XP Beads (1.0X). The ligation products were then amplified by polymerase chain reaction (PCR). The resulting cDNA library was sequenced using the Illumina Novaseq6000 platform by Gene Denovo Biotechnology Co. (Guangzhou, China).

### 4.4. Sequencing Data Processing

To obtain high-quality clean reads, raw reads from the sequencing machines were filtered using fastp [51] (version 0.18.0). The filtering parameters included removing reads containing adapters, reads with more than 10% unknown nucleotides (N), and low-quality reads with more than 50% of bases having a Q-value ≤ 20. The short-read alignment tool Bowtie2 [52] (version 2.2.8) was used to map reads to the ribosome RNA (rRNA) database. Reads mapped to rRNA were subsequently removed. The remaining clean reads were further used for assembly and gene abundance calculation. An index of the reference genome was constructed, and paired-end clean reads were mapped to the reference genome using HISAT2 v2.4 [53] with default parameters. The mapped reads of each sample were assembled using StringTie v1.3.1 [54,55] in a reference-based approach. For each transcription region, the FPKM (fragments per kilobase of transcript per million mapped reads) value was calculated using RSEM [56] to quantify expression abundance and variations.

### 4.5. Sequencing Data Analysis

Differential expression analysis of RNAs was conducted using DESeq2 [57] software between two different groups. Genes with a false discovery rate (FDR) below 0.05 and an absolute fold change ≥ 2 were identified as differentially expressed genes (DEGs). Pathway enrichment analysis was performed using the public pathway-related database KEGG [58]. Protein–protein interaction (PPI) networks were constructed using STRING v10 [59], with genes represented as nodes and interactions as edges in the network. The network was visualized using Cytoscape (v3.7.1) [60] software to identify core and hub genes and their biological interactions. Weighted gene co-expression network analysis (WGCNA) was conducted using OmicShare tools, a free online platform for data analysis (https://www.omicsmart.com/WGCNA/, accessed on 10 May 2024). To ensure high reliability, the parameters were set as follows: power value = 6, minimum number of genes per module = 50, maximum number of modules = 20, and retention expression = 10. Gene significance (GS) and module membership (MM) were calculated to correlate the modules with the phenotypic data (ABA, JA, IAA, ZT, and GA_3_ activities).

### 4.6. qRT-PCR Analysis

Samples identical to those used for RNA-seq were prepared for qRT-PCR analysis. Total RNA isolated from RNA-seq samples was used to synthesize first-strand cDNA using the PrimeScript™ RT reagent Kit (Takara, Dalian, China). The 2 × SG Fast qPCR Master Mix (BBI, Shanghai, China) was used to detect gene expression patterns, with *GAPDH* (NCBI accession No. EF189713.1) serving as the reference gene. Primers used are listed in Appendix A. The 2^−∆∆CT^ relative quantification method was used to calculate the relative gene expression. Three biological replicates were included per sample, and standard errors were calculated based on these replicates.

### 4.7. Statistical Analysis

ANOVA of phytohormone activities and calculation of standard errors were performed using SPSS 21 (Chicago, IL, USA). A *p*-value ≤ 0.05 was considered statistically significant. The data were presented as mean ± standard error (SE) from three biological samples.

## 5. Conclusions

Taken together, under low-temperature stress, IBA enhanced the activities of phytohormones in sugarcane roots, primarily by inducing the expression of the *CICDH*, *GSTF11*, *GSTU6*, *GST1*, *GST4*, *BCAT3*, *HIBCH*, *FAR1*, *FAR2*, *FAR5*, *AGAL1*, *Ub-CEP52-1*, *PRL19A*, *RPS28*, *RPS25B*, *ACS1*, *RPP3A*, *RPS3C*, and *ACC1* genes, thereby regulating the pathways of glutathione metabolism; peroxisome; cutin, suberine and wax biosynthesis; glucosinolate biosynthesis; valine, leucine and isoleucine degradation; and glycosphingolipid biosynthesis (globo and isoglobo series). Consequently, sugarcane roots overcame low-temperature stress, leading to enhanced root system development (Figure 7). This study offers novel insights into the molecular mechanisms through which IBA promotes root development under low-temperature conditions, providing a valuable theoretical basis for improving cold resilience in sugarcane and other crops.

## Figures and Tables

**Figure 1 plants-14-01502-f001:**
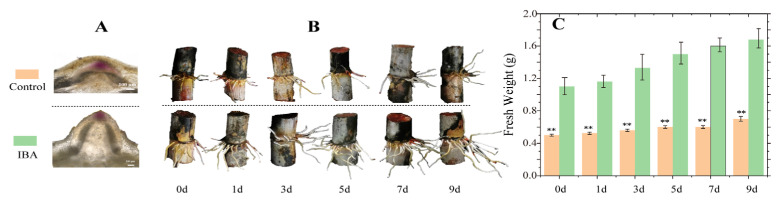
The effect of IBA on sugarcane root development under low-temperature conditions. (**A**) Root primordia formation in seedcane at 15 days after the first IBA treatment. (**B**) Root system development in seedcane at 0, 1, 3, 5, 7, and 9 days after the second IBA treatment. (**C**) Fresh root weight of seedcane at 0, 1, 3, 5, 7, and 9 days after the second IBA treatment. The orange bars represent the control treatment, while the green bars represent the IBA treatment. Asterisks (**) denote a statistically significant difference (*p* < 0.01) between the control and IBA treatments at each time point.

**Figure 2 plants-14-01502-f002:**
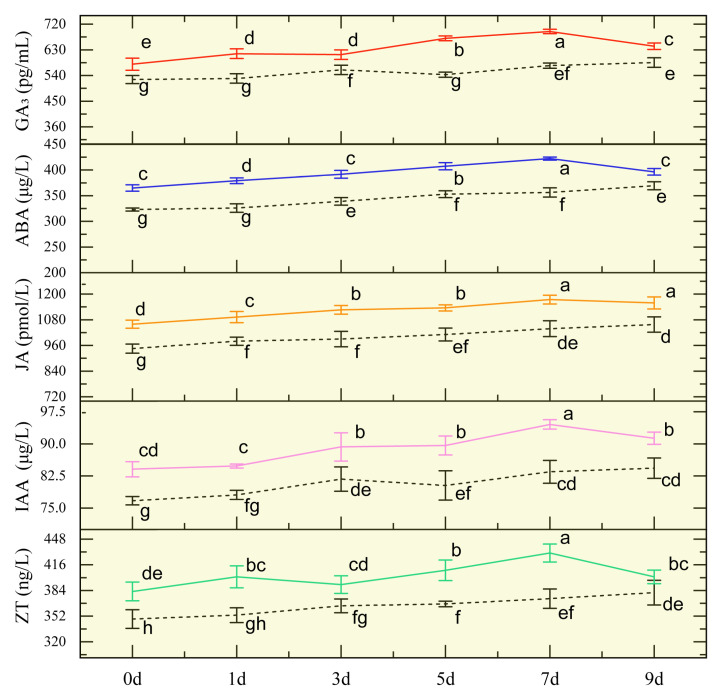
Phytohormone activities in seedcane roots under IBA treatment at low temperature. Means labeled with different letters indicate statistically significant differences (*p* < 0.05) based on Duncan’s multiple range test. Solid lines in red, blue, orange, pink, and green, from top to bottom, represent the content variation of GA_3_, ABA, JA, IAA, and ZT under IBA treatment, respectively. Black dashed lines, from top to bottom, represent the content variation of GA_3_, ABA, JA, IAA, and ZT under control conditions, respectively.

**Figure 3 plants-14-01502-f003:**
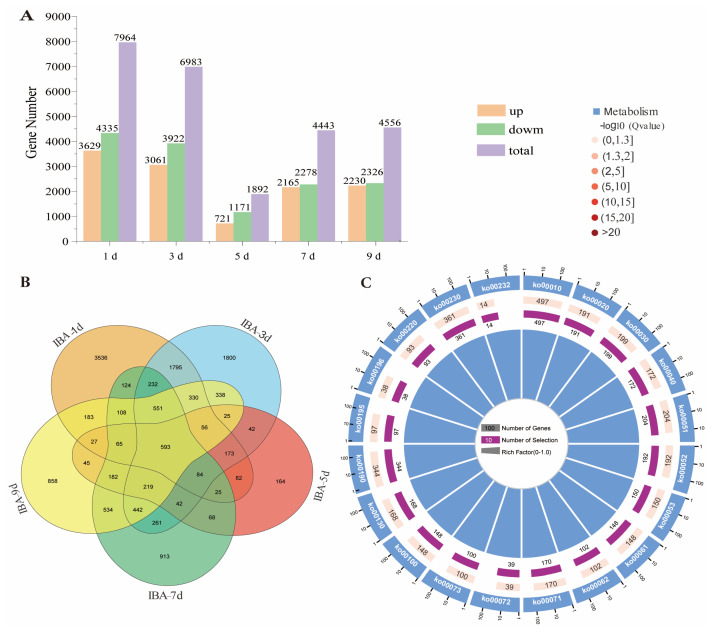
Identification of DEGs induced by IBA under low temperature. (**A**) IBA treatment induced a significant number of responsive genes in sugarcane roots. (**B**) DEGs uniquely and consistently expressed at 1, 3, 5, 7, and 9 days after IBA treatment. (**C**) KEGG pathway enrichment analysis of DEGs identified across all time-point comparisons. From outer to inner rings: the outermost ring shows the DEG number scale; the first ring represents the top 20 enriched pathways, each bar indicating one KEGG pathway ID; the second ring displays the number of background genes enriched in each pathway and corresponding Q-values (darker bars represent smaller Q-values and longer bars indicate higher background gene counts); the third ring shows the gene ratio of DEGs enriched in each pathway (all bars in dark purple, with specific values labeled); the innermost ring indicates the Rich Factor values for each pathway.

**Figure 4 plants-14-01502-f004:**
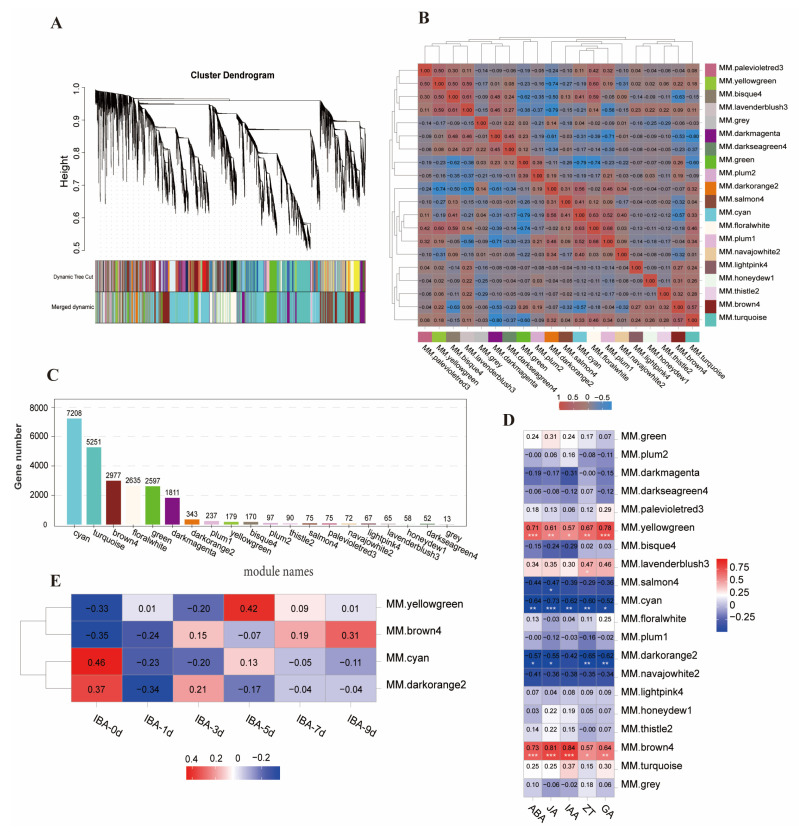
Weighted gene co-expression network analysis (WGCNA) of DEGs across all time points. (**A**) Clustering of genes into modules based on similar expression profiles. (**B**) Heatmap of pairwise module correlations, where darker colors indicate stronger correlations. (**C**) Distribution of gene counts across the identified modules. (**D**) Correlation between modules and phytohormone levels (ABA, JA, IAA, ZT, and GA_3_). (**E**) Temporal expression patterns of phytohormone-associated modules, correlation coefficients are color-coded based on their scores. One, two, and three asterisks denote significance at * *p* < 0.05, ** *p* < 0.01, and *** *p* < 0.001, respectively.

**Figure 5 plants-14-01502-f005:**
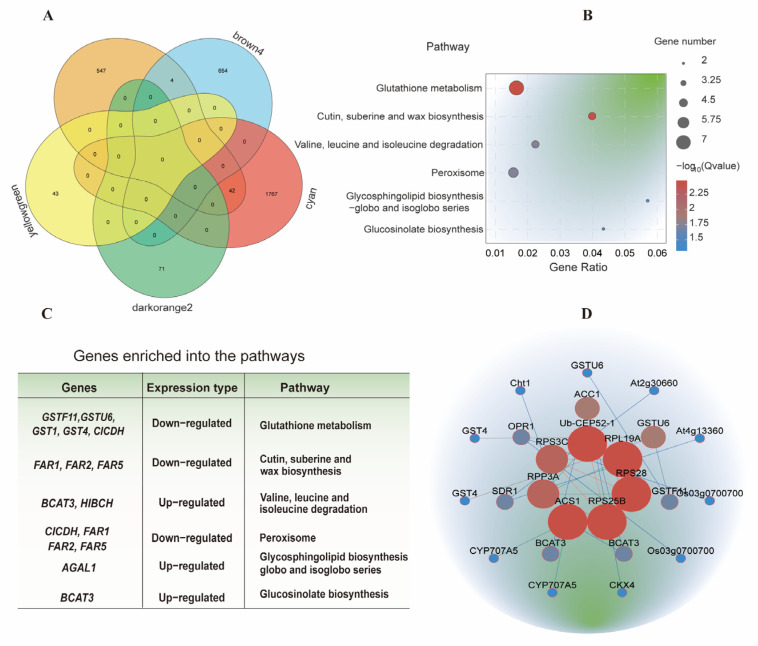
Identification of core genes associated with IBA stimulation under low-temperature conditions. (**A**) Venn diagram analysis of common genes identified from all DEGs and WGCNA-derived gene modules. (**B**) KEGG pathway analysis of core genes responsive to IBA stimulation in sugarcane roots under low-temperature conditions. (**C**) Candidate genes involved in IBA stimulation under low-temperature conditions. (**D**) PPI network highlighting hub genes associated with phytohormone activities under low-temperature conditions.

**Figure 6 plants-14-01502-f006:**
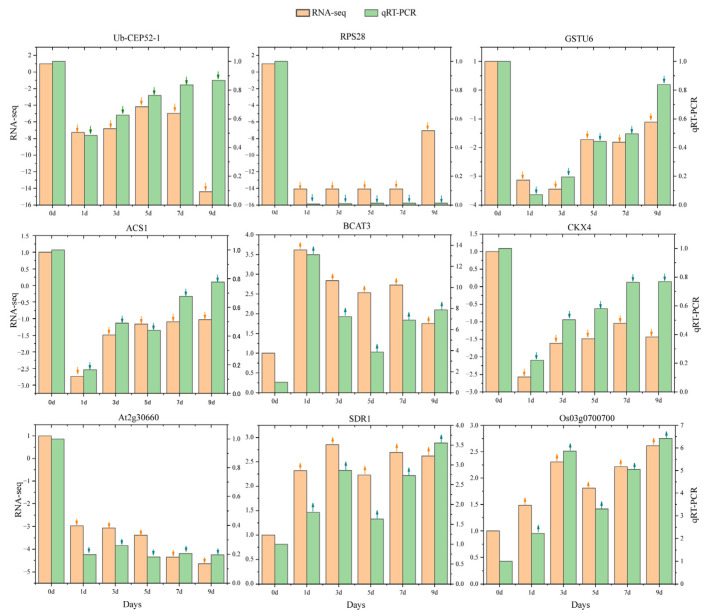
qRT-PCR validation of core gene expression patterns in response to IBA stimulation under low temperature. The annotation above each bar chart corresponds to gene names, with the left y-axis scale indicating gene expression levels quantified by RNA-seq, and the specific values at different time points shown by orange bars. The right y-axis scale displays expression values measured via qRT-PCR, with specific values at different time points shown by green bars. ↑: upregulated; ↓: downregulated.

**Figure 7 plants-14-01502-f007:**
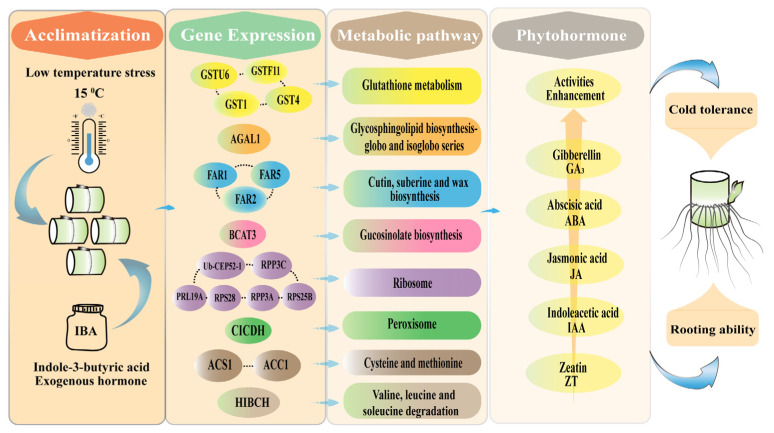
Proposed mechanism of sugarcane root response to IBA stimulation under low-temperature stress. IBA stimulation alters the expression of key genes, including *CICDH*, *GSTF11*, *GSTU6*, *GST1*, *GST4*, *BCAT3*, *HIBCH*, *FAR1*, *FAR2*, *FAR5*, *AGAL1*, *Ub-CEP52-1*, *PRL19A*, *RPS28*, *RPS25B*, *ACS1*, *RPP3A*, *RPS3C*, and *ACC1*, thereby modulating critical pathways such as glutathione metabolism; peroxisome function; cutin, suberin, and wax biosynthesis; glucosinolate biosynthesis; valine, leucine, and isoleucine degradation; and glycosphingolipid biosynthesis (globo and isoglobo series). These regulatory changes enhance phytohormone activities, enabling sugarcane roots to better tolerate low-temperature stress and promoting robust root system development.

## Data Availability

All the data supporting the findings of this study are available within the article and its Appendix A. The raw data have been deposited in NCBI’s database and are accessible through accession number PRJNA1109933 (https://www.ncbi.nlm.nih.gov/sra/?term=PRJNA1109933 (accessed on 10 August 2024)).

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
