# Peer review of "Indole-3-Butyric Acid Enhances Root Formation and Alleviates Low-Temperature Stress in Sugarcane: Molecular Insights and Identification of Candidate Genes"

_plants, 2025, doi:10.3390/plants14101502_

Round 1

Reviewer 1 Report

Comments and Suggestions for Authors

Legends of the Figures 1, 2 and 6 need to improve with more details. The schematic representation of the figure 7 would be made in BioRender and check the correct terminology.

Author Response

70-05-2025

Dear reviewer,

Many thanks for your positive feedback. Your suggestions and comments have greatly contributed to improving our manuscript. We have incorporated the suggestions into the revised manuscript, and we hope that you will find the revisions satisfactory.

Legends of the Figures 1, 2 and 6 need to improve with more details. The schematic representation of the figure 7 would be made in BioRender and check the correct terminology.

Thank you. Figure 1, 2, 7 have been updated according to your suggestions.

We look forward to hearing a positive reply from you.

Kind regards,

Dong-Liang Huang, PhD

Professor, Guangxi Academy of Agricultural Sciences, Nanning, Guangxi, China

Reviewer 2 Report

Comments and Suggestions for Authors

Abstract
Please keep your abstract short. The publisher's requirement is 200 words.

Keywords
I suggest changing the keywords so that they do not repeat with words from the title.

introduction
Line 75-80 contains the results of the study. Please move them to the results chapter. The introduction does not describe the results obtained.

Resoults
No comment

Discussion
No comment

Material and methodology
Line 321 Were the plants incubated in a culture room with controlled conditions? Information on light output can be added. Were they in the same place throughout the experiment?
Line 336 Were the laboratory procedures developed by the authors of the article? If not please provide literature sources describing these procedures. 
Line 348 same

References
No comment

Author Response

07-05-2025

Dear reviewer,

Many thanks for your positive feedback. Your suggestions and comments have greatly contributed to improving our manuscript. We have incorporated the suggestions into the revised manuscript, and we hope that you will find the revisions satisfactory.

Abstract
Please keep your abstract short. The publisher's requirement is 200 words.

Thank you. The Abstract has now been revised to under 200 words.

Keywords
I suggest changing the keywords so that they do not repeat with words from the title.

Thank you. The key words have been updated.

introduction
Line 75-80 contains the results of the study. Please move them to the results chapter. The introduction does not describe the results obtained.

Thank you. This section has been revised in accordance with your suggestions.

Material and methodology
Line 321 Were the plants incubated in a culture room with controlled conditions? Information on light output can be added. Were they in the same place throughout the experiment?

Yes, further details have been included in this section, described as follows: “Single-bud seedcanes, approximately 5 cm in length, were immersed in an IBA solution (60 mg/L) and then incubated at a constant temperature of 15 °C under a 16 h light/8 h dark photoperiod with a light intensity of 2000 Lux. The seedcanes were maintained in a fixed location throughout the experiment”.

Line 336 Were the laboratory procedures developed by the authors of the article? If not please provide literature sources describing these procedures.

Yes, we have included this information as follows: “The phytohormones were extracted according to the method described by Li et al. [51], with slight modifications.”

Line 348 same

“This procedure was performed following the protocol provided by Gene Denovo Biotechnology Co. (Guangzhou, China)”. This information has been included in this section.

We look forward to receiving a positive reply from you.

Kind regards,

Dong-Liang Huang, PhD

Professor, Guangxi Academy of Agricultural Sciences, Nanning, Guangxi, China